# Molecular Epidemiology Analysis of SARS-CoV-2 Strains Circulating in Romania during the First Months of the Pandemic

**DOI:** 10.3390/life10080152

**Published:** 2020-08-14

**Authors:** Marius Surleac, Leontina Banica, Corina Casangiu, Marius Cotic, Dragos Florea, Oana Sandulescu, Petre Milu, Anca Streinu-Cercel, Ovidiu Vlaicu, Dimitrios Paraskevis, Simona Paraschiv, Dan Otelea

**Affiliations:** 1“Prof. Dr. Matei Bals” National Institute for Infectious Diseases, 021105 Bucharest, Romania; surleac@biochim.ro (M.S.); leontina.banica@gmail.com (L.B.); mariuscotic@yahoo.com (M.C.); dragos.florea@umfcd.ro (D.F.); oana.sandulescu@umfcd.ro (O.S.); anca.streinucercel@umfcd.ro (A.S.-C.); vlaicu.ovidiu@yahoo.com (O.V.); dotelea@mateibals.ro (D.O.); 2Institute of Biochemistry, Romanian Academy, 060031 Bucharest, Romania; 3“Marie Curie” Emergency Clinical Hospital, 077120 Bucharest, Romania; corina.casangiu@gmail.com; 4“Carol Davila” University of Medicine and Pharmacy, 050474 Bucharest, Romania; petre.milu@gmail.com; 5National Retrovirus Reference Center, Department of Hygiene and Epidemiology, Faculty of Medicine, National and Kapodistrian University of Athens, 15772 Athens, Greece; dparask@med.uoa.gr

**Keywords:** SARS-CoV-2, WGS, coronavirus, phylogenetic analysis, outbreak, mutations, pandemic, Romania

## Abstract

BACKGROUND: The spread of SARS-CoV-2 generated an unprecedented global public health crisis. Soon after Asia, Europe was seriously affected. Many countries, including Romania, adopted lockdown measures to limit the outbreak. AIM: We performed a molecular epidemiology analysis of SARS-CoV-2 viral strains circulating in Romania during the first two months of the epidemic in order to detect mutation profiles and phylogenetic relatedness. METHODS: Respiratory samples were directly used for shotgun sequencing. RESULTS: All Romanian sequences belonged to lineage B, with a different subtype distribution between northern and southern regions (subtype B.1.5 and B.1.1). Phylogenetic analysis suggested that the Romanian epidemic started with multiple introduction events from other European countries followed by local transmission. Phylogenetic links between northern Romania and Spain, Austria, Scotland and Russia were observed, as well as between southern Romania and Switzerland, Italy, France and Turkey. One viral strain presented a previously unreported mutation in the Nsp2 gene, namely K489E. Epidemiologically-defined clusters displayed specific mutations, suggesting molecular signatures for strains coming from areas that were isolated during the lockdown. CONCLUSIONS: Romanian epidemic was initiated by multiple introductions from European countries followed by local transmissions. Different subtype distribution between northern and southern Romania was observed after two months of the pandemic.

## 1. Introduction

The human population has been exposed to more than 50 emerging (or re-emerging) infectious diseases during the past half-century; 75% of them were vector-borne or zoonotically transmitted [1]. The rapid spread of these emerging infectious diseases was facilitated by increased international travel as was the case with the coronavirus infectious disease 2019 (COVID-19) pandemic.

A new variant of coronavirus, the Severe Acute Respiratory Syndrome Coronavirus 2 (SARS-CoV-2) emerged from East Asia (Wuhan, China) at the end of 2019 [2] and spread across the world. SARS-CoV-2 causes a wide spectrum of clinical manifestations with different severities in humans, ranging from asymptomatic to life-threatening critical cases [3,4]. Romania reported the first confirmed case of SARS-CoV-2 on February 26th, 2020 [5]. Since then, 28973 new cases and 1750 COVID-19 related deaths were reported (https://gisanddata.maps.arcgis.com/ accessed on 6th of July). It has been shown that SARS-CoV-2 can be transmitted by both symptomatic and asymptomatic individuals via micro-droplets [6]. Many countries including Romania adopted a period of state of emergency to flatten the curve of infections and to slow down the spread of the virus. In Romania, it was instituted from the 16th of March to the 14th of May and was followed by a state of alert in which restricted circulation and social distancing rules were applied to varying degrees. An important number of Romanian citizens (~5 million) have been and are currently working abroad in South-West European countries such as Italy, Spain, UK, France, Germany that were highly affected by the COVID-19 pandemic. There is a significant amount of crisscrossing between these countries and Romania. Several outbreaks were reported in different geographical parts of Romania; Bucharest metropolitan area (BMA) and Suceava county were the most affected regions.

Epidemiological surveillance of circulating viruses enhanced by analyses of viral genetic information has proven useful in tracing the origin and spread of several viruses (Ebola, Dengue, HIV) [7,8,9]. Early molecular epidemiology studies on SARS-CoV-2 revealed the initial zoonotic transmission of this virus from bats (the main reservoir of coronaviruses) and possibly other intermediate hosts (e.g., Malaya pangolins) [10,11]. Although the evolutionary rate of SARS-CoV-2 is lower than that observed in other RNA viruses, probably due to the proofreading function of the RNA polymerase, the viral strains are accumulating mutations and evolving continuously. Three main genetic groups of SARS-CoV-2 have been described: A, B, C, each with several subtypes [12]. International scientific teams have successfully collaborated during the COVID-19 pandemic, generating an unprecedented number of viral genome sequences and making them available through open platforms such as GISAID (www.gisaid.org). On this platform, the data about the ongoing evolution of the virus is continuously updated.

Our goal was to analyze the viral variants circulating in Romania by using molecular phylogeny tools in order to better understand the dynamics and the dispersal routes of this virus, during the first months of the epidemic.

## 2. Materials and Methods 

### 2.1. Study Population

Twenty-five respiratory samples from SARS-CoV-2 infected patients were selected for viral sequencing using a shotgun strategy. The COVID-19 diagnosis was established by RT-PCR tests (Coronavirus (COVID-19) CE IVD, Genesig, UK). Selection criteria included: main Romanian outbreak representativeness (Suceava county, BMA), high viral load samples (Ct values lower than 25), expanded time frame sample collection (the first two months of the epidemic). Before sequencing, the respiratory samples were stored at −70 °C. The patients with viral strains sequenced in this analysis displayed the whole spectrum of clinical presentations.

### 2.2. Whole-Genome Sequencing (WGS) of SARS-CoV-2 Strains

The viral RNA extraction was performed using QIAamp® DSP Virus Kit (Qiagen, Hilden, Germany). The protocol was performed as recommended by the manufacturer with one additional step, the treatment with RNase-Free DNase Set (QIAGEN). The quality of the extracted RNA was evaluated with Agilent RNA 6000 Nano Kit on 2100 Bioanalyzer and low contamination with ribosomal RNA (rRNA) was observed. Viral extraction was followed by DNA library preparation that was conducted with TruSeq Stranded Total RNA kit (Illumina, San Diego, CA, USA) without the rRNA depletion procedure. Quantification of DNA libraries was done using Qubit™ dsDNA HS Assay Kit and the Qubit4 Fluorometer (Thermo Fisher Scientific, Waltham, Massachusetts, United States). The quality of the DNA libraries was determined by Agilent 2100 Bioanalyzer using High Sensitivity DNA kit. The average size of the libraries was 260 bp. Finally, DNA libraries consisting of 4−5 samples were sequenced using Illumina® MiSeq® Reagent Kit v3 with a number of sequencing cycles equal to 150 and a load concentration of 10 pM per library.

### 2.3. WGS Assembling and Reference Mapping

The generation of the consensus sequences follows a double-assembly method approach in which multiple rounds of mapping combined with de novo assembling were used. The paired-end (PE) raw reads were first de novo assembled with SPAdes 3.12.0 implemented in shovill (https://github.com/tseemann/shovill) [13]. Prior to this analysis, the raw data were prepared using trimming tools already implemented in shovill in order to remove the adapters and correct the sequencing and assembly errors. The resulting de novo contigs for each sample were then uploaded to Geneious Prime® 2020.1.2. software and then mapped onto a database of over 5000 NCBI SARS-CoV-2 sequences (implemented in Geneious Prime). In the mapping process of the contigs set in Geneious Prime we used a quality filter of >Q30, with High Sensitivity settings and Fine Tuning (up to 5 iterations). The de novo contigs representing the viral full genome had an average coverage of 182 reads with a minimum of 16 and the maximum of 543 reads.

MegaBLAST [14] was used to extend the reference sequence search. At this step, a first consensus sequence was generated for each sample, based on all the reference sequences used during the mapping process. This step was further refined by re-mapping the contigs to the particular reference sequence with the highest number of mapped contigs or re-mapping on more than one reference. These steps generated one or more consensus sequences. For most of the samples, de novo assembly generated a set of contigs of which the longest one (approximately 29k nucleotides) represented the whole genome of SARS-CoV-2. PE raw reads were further uploaded in Geneious Prime and mapped on one or more reference sequence hits found previously in mapping de novo contigs. The mapping of the raw reads was performed with similar settings in Geneious Prime. One or more consensus sequences were obtained. This step was implemented in our analysis to validate the correctness of WGS assembly. Finally, one consensus sequence corresponding to the viral genome was obtained for each sample.

### 2.4. Phylogenetic Analysis

The analyzed Romanian sequences were aligned along with control sequences using a multiple sequence alignment tool (MSA). MAFFT v7.450, implemented in Geneious Prime, was used [15] with the following settings: FFT-NS-i × 1000: Medium iterative refinement method, two cycles only; 100PAM/k = 2 scoring matrix; gap open penalty: 1.53.

The selection of the control sequences was done by performing BLAST search in both GISAID (https://www.gisaid.org) and GenBank (https://www.ncbi.nlm.nih.gov/genbank/) platforms for each of the 25 SARS-CoV-2 genomes generated in this study. Twenty six other Romanian sequences were retrieved from GISAID (22 sequences from Laboratory for Respiratory Viruses, National Influenza Centre, Cantacuzino National Military-Medical Institute for Research and Development, Bucharest, Romania; and 4 other sequences from National Influenza Centre Romania, Bucharest, Romania & Charite Universitatsmedizin Berlin, Institute of Virology, Berlin, Germany). Due to hyper mutation scores, only 24 of these were kept in the final data set. The BLAST search in the two databases mentioned above returned a total of 346 sequences most similar to the Romanian sequences. A set of 40 outgroup sequences was also selected from the GISAID platform (20 sequences for each of the SARS-CoV-2 A and C types/lineages), as classified in a recent study [12].

The 435 sequences in the final data set were aligned with MAFFT and the alignment was used afterwards as input for the phylogenetic analysis. The aligned sequences were trimmed at the “gapped”-ends (due to the presence or absence of nucleotides in these regions as a result of sequencing). The phylogenetic trees were generated with RAxML 8.2.11 [16], Nucleotide model: GTR GAMMA I, Algorithm: Rapid hill-climbing, Number of starting trees or bootstrap replicates: 1, Parsimony random seed: 1.

### 2.5. Lineage and Mutations Prediction

All of the Romanian SARS-CoV-2 sequences were subject to lineage, amino acid and nucleotide mutation predictions. The following online tools were used for these purposes: Pangolin COVID-19 Lineage Assigner—https://pangolin.cog-uk.io [17]; Covidex using Rambaut classification model—https://cacciabue.shinyapps.io/shiny2/ [17]; CoV-GLUE—http://cov-glue.cvr.gla.ac.uk/#/home [17,18].

### 2.6. Variability Profiling

Variability analysis on WGS of SARS-CoV-2 was performed using an in-house Python script on two data sets: the 25 SARS-CoV-2 sequences analyzed in this study and all 51 SARS-CoV-2 Romanian samples available in the GISAID database. This script uses three nucleotide-specific substitution matrices (e.g., Needleman-Wunsch, Todd Lowe, HoxD70) and generates a consensus score for each alignment as well as a substitution score for each nucleotide in the alignment. An average rescaled variability consensus is finally obtained. The script provides a position-based variability profile as well as a variability profile for each full sequence in an alignment.

### 2.7. Graphic Representations

The figures depicting the distribution of mutations within our data set were created with BioRender.com. For visualization of phylogenetic trees, we used FigTree v 1.4.4 software. The 3D models of SARS-CoV-2’s Nsp2 and Nsp4 were generated with UCSF Chimera 1.14 visualization and analysis of molecular structures software [19].

### 2.8. Nucleotide Sequence Accession Numbers

The WGS sequences generated in this study were submitted to the GISAID platform (under the accession numbers EPI-ISL-468134 to EPI-ISL-468158). 

## 3. Results

### 3.1. Phylogenetic and Subtyping Analysis

The subtype analysis of the 25 SARS-CoV-2 sequences generated in this study indicated that all Romanian sequences belong to lineage B. Furthermore, all the SARS-CoV-2 sequences corresponding to viral strains circulating in the BMA were classified as subtype B.1.1, whereas the sequences from the Suceava county outbreak were assigned as B.1.5.

We have performed a phylogenetic analysis of the Romanian SARS-CoV-2 sequences generated in this study in comparison with the reference sequences available on the GISAID platform and the GenBank nucleotide database. The most closely related sequences to the Romanian data were selected as references. The resulting phylogenetic tree is presented in Figure 1. Twenty-four other Romanian sequences retrieved from GISAID were also included in the analysis. All of the analyzed Romanian sequences (49 in total) were marked with red in the tree. An additional set of 40 sequences (type A in orange and type C in green) were used as outgroups. As it can be seen in Figure 1, Romanian sequences are dispersed throughout the B type lineage (highlighted as light yellow in the tree). Several of our sequences have formed clusters which also incorporated sequences from different parts of the world. This may suggest that virus introduction in Romania was followed by local transmission events or that identical strains have been introduced from different geographic locations. For the sequences presented as single branches, they provide single introduction events. Overall, the SARS-CoV-2 circulating strains in Romania are the result of multiple introductions, mainly from other European countries. Romanian sequences formed several region-specific clusters, highlighted in the phylogenetic tree.

The sequences corresponding to viral strains circulating in BMA were grouped in two different clusters (Bucharest cluster 1 and 2). The Bucharest cluster 1 sequences (Figure 1 pink highlight) were most closely related to sequences from Switzerland, Italy, France, Turkey and the USA. Bucharest cluster 2 (Figure 1 blue highlight) consists of closely related sequences which were all generated using samples originating from infected healthcare personnel working at a single medical institution. Both BMA clusters group with other European sequences (Germany, Spain) in a cluster of type B strains (subtype B.1.1). The sequences corresponding to viral strains circulating in Suceava County (Figure 1 purple highlight) clustered together with sequences from Spain (Barcelona, Valencia), Russia (St. Petersburg), Austria, Scotland and Sweden. However, the Suceava County sequences do not display tight clustering, suggesting multiple introductions in this particular geographic area.

### 3.2. Mutations Analysis

The mutation analysis of the Romanian full-genome sequences indicated that the frequency of synonymous and non-synonymous substitutions is similar, with a slight increase in the non-synonymous (56.6%, n = 69) compared to the synonymous (43.4%, n = 53). We noticed a difference in the non-synonymous mutational frequencies when analyzing the viral genomes sequenced in this study and other Romanian sequences available in GISAID platform: ~25% versus ~75% of the total non-synonymous mutations accounted. Figure 2 illustrates the mutated sites along the viral genome in the analyzed sequences. The analysis indicated several mutations that were linked to lineage B of SARS-CoV-2. These mutations, C3037T, C14408T (P323L in Nsp12) and A23403G (D614G in S), were present in all Romanian sequences. When looking into the GISAID database mutational statistics, which uses hCoV-19/Wuhan/WIV04/2019 EPI_ISL_402124 as a reference strain (COVsurver tool), both P323L and D614G mutations are present in a high percentage (61.7%/61.8%, n = 33480 and n = 33537).

The highest diversity of mutations was found in clusters from the South of Romania. The sequences from the BMA presented specific mutations: G28881A, G28882A (R203K in N gene) and G28883C (G204R in N gene). These mutations were also observed in other sequences collected from the South of Romania (Figure 2), probably being associated with subtype B.1.1. The frequency of R203K and G204R variants in the GISAID database mutational statistics is 23.6% (n = 12,784). Interestingly, in the sequences from Bucharest cluster 1 (Figure 1 pink highlight), we observed a genetic signature, a synonymous mutation (T19839C). Furthermore, all of the sequences belonging to Bucharest cluster 2 (Figure 1 blue highlight) presented a particular non-synonymous mutation, C16049T (T870I in Nsp12). One of these sequences had additional mutations: A9744G (Y397C in Nsp4), A22803C and C28603T. The patient infected with this viral strain had a severe form of COVID-19, requiring management in intensive care.

All of the sequences belonging to the Suceava cluster (Figure 1 purple highlight) presented a particular synonymous mutation, A20268G that is related to subtype B.1.5 assignment. Subtyping predictions made with the Pangolin webserver show that all the sequences having this synonymous mutation belong to B.1.5 subtype. This is a possible outbreak-defining signature of SARS-CoV-2 circulating strains in this region, given that Suceava County was an isolated region, quarantined throughout most of the emergency state period.

Of particular interest is the K489E mutation, found in one of the Romanian sequences SARS-CoV-2 sequence from Bucharest cluster 2 (EPI_ISL468150) and tagged with a black star on top of the mutation marker in Figure 3. The patient infected with this viral strain presented mild symptoms. This is the first study to report this mutation found in the Nsp2 protein-coding gene of SARS-CoV-2. Yet, there is no 3D solved crystal structure for this non-structural protein but the team behind C-I-TASSER pipeline have already generated 3D models for SARS-CoV-2 proteins (https://zhanglab.ccmb.med.umich.edu/COVID-19/) and when looking closer into the structure of Nsp2 (ID-QHD43415_2.pdb) it seems that this amino acid (K489) is involved in an H-bond interaction with the oxygen from the main chain of L553. At the same time, another residue from Nsp2, namely K557 is connected to the same oxygen atom from L553 through an H-bond. Therefore, the mutation to a glutamic acid may disrupt the structural conformation in that region by breaking those two H-bonds and forming a salt-bridge with K557. If this region is involved in any kind of interaction with other proteins, then this mutation could be highly important. There are only two other different mutations in the GISAID database at this particular position in Nsp2, but none to describe a K-to-E mutation. The other two mutations in this position are to N/R (EPI_ISL_448990 and EPI_ISL_478285, respectively). Depending on the secondary structures, the surrounding or/and neighbouring amino acids or even if the amino acid is on the surface or in the core of the protein, there are diverse preferences between the amino acids involved in salt-bridges, with a slight preference of Lys towards Glu (as in our case).

Substitutions implying changes into Thymine were most frequently observed in the analyzed Romanian sequences, for both non-synonymous (38%, n = 46) and synonymous (23%, n = 28) mutations. There is a slight preference between the two groups when mutating to the other nucleotides: Adenine is the next most abundant in the non-synonymous group (8.2%, n = 10) while Cytosine is the next most abundant in the synonymous group (9.8%, n = 12).

When analyzing all the available Romanian sequences in GISAID platform, the lineage predictions show that the mutations were almost equally distributed between B1 (34.7%, n = 17), B.1.1 (46.9%, n = 23) and B.1.5 (38.8%, n = 19) subtypes. We also observed mutational signatures for several of the country regions where samples were collected (Appendix A).

### 3.3. Variability Profile

The variability of the analyzed SARS-CoV-2 Romanian sequences, when compared to each other, showed that most of the genomes were highly conserved (Appendix A). The most conserved sequences were mostly part of the Suceava cluster (highlighted in purple in the tree). The group of sequences from Bucharest has a slightly increased variability, having also the highest number of mutations (at the amino acid and nucleotide level). The two most variable sequences (one from each group) had a variability score of 6.3 (Bucharest) and 6.9 (Suceava) on a scale from 0 to 9. The sequence from Suceava County with the highest variability score had 11 nucleotide mutations translated into 3 amino acid mutations; the other most variable sequence from Bucharest had 7 nucleotide mutations translated into 5 amino acid mutations. When looking in the genome at the variability for each nucleotide position, we found high variability scores at the following positions: 2593 (score = 3), 16,049 (score = 4), 19,839 (score = 4), 20,268 (score = 9) and 28,881/28,882/28,883 (score = 9/9/7).

## 4. Discussion

The new human-infecting coronavirus, SARS-CoV-2, circulating in late December 2019 in Wuhan, China, was identified using NGS [2]. In Europe, the first country affected by the new coronavirus was Italy, followed by Spain, Germany, France, and the UK. It is estimated that up to 6 million Romanian citizens are working abroad, mainly in these countries. 

As the COVID-19 epidemic started accelerating, healthcare and economic uncertainties persuaded many Romanian citizens working in these countries to return home. According to the Romanian authorities, between the end of February and the beginning of May, almost 1,279,000 citizens returned home from these countries [20]. In Romania, the first case of SARS-CoV-2 infection was registered at the end of February. During the first six weeks of the epidemic in Romania, nearly half of the SARS-CoV-2 cases were registered in persons returning from other European countries (mainly Italy, followed by Spain and UK) or in those in close contact with them [21]. Two months after the first case was registered, outbreaks were present in multiple geographic regions; the highest number of cases was reported in Suceava county (n = 3453) followed by Bucharest metropolitan area (n = 2321), representing 18.9% and 12.7%, respectively, of all national cases [22]. The highest density of cases was registered in Suceava county (45.19 cases per 10,000 habitants) suggesting significant local transmission from undetected imported cases [22]. Suceava was the first county in Romania to report community transmission of SARS-CoV-2, with a sudden outbreak of severe COVID-19 cases in patients with no apparent travel history, in the second half of March 2020, when the county’s case count jumped in only two weeks from 8 imported/close contact cases (19 March 2020) to 701 community-acquired or nosocomial (healthcare workers) cases (2 April 2020). The city of Suceava and 8 surrounding towns were placed under quarantine from 30 March 2020 to 13 May 2020, with the number of new cases per day in this span ranging from 2 to 248, with a median of 47 (IQR: 25−94) new cases and with a fluctuating, slowly decreasing pattern (Appendix A). In the same period, in Bucharest, the daily new case count ranged from 2 to 76 (median 23, IQR: 16−35).

The phylogenetic analysis performed on the viral sequences collected during the first period of the Romanian epidemic suggested a clear segregation between the North of the country (Suceava County) where subtype B.1.5 was exclusively present and the South, where subtype B.1.1 was present. The segregation of particular subtypes was observed probably as a result of the lockdown measures applied during the two-month state of emergency and also that the two epidemics were founded by different strains. To sustain this hypothesis, our results showed that the samples from Suceava county (collected two months after the beginning of the outbreak) were still assigned as subtype B.1.5 and localized in the same cluster with the samples from the start of the epidemic. This might change after lifting the lockdown measures. New outbreaks were reported last month in other Romanian regions (e.g., Vrancea, Brasov, Buzau).

The sequences of the viral strains circulating in Suceava county were closely related to samples collected in Spain (Barcelona, Valencia), Austria, Scotland, Russia (St. Petersburg). Most likely, the index cases were imported from other European countries and the virus initially circulated undetected, leading to local community transmission, until a large outbreak was registered in an emergency hospital in Suceava. Here, an important number of healthcare workers acquired the infection from a super-spreader patient with community-acquired pneumonia, who did not initially meet the national case definition for COVID-19 which, up to 23 March 2020, required an epidemiological context in order to necessitate testing for SARS-CoV-2 (i.e., travel history to a list of specific countries, or contact with a confirmed case) [23]. After 23 March 2020, when the national surveillance methodology was updated to include testing for SARS-CoV-2 in cases of severe acute respiratory tract infection (SARI) of unknown aetiology [24], a high number of cases of COVID-19 were detected and reported in Suceava, with an important rate of severe and critical cases. The variability of viral strains from Suceava was lower when compared with BMA, suggesting that the circulating viruses were seeded from a lower number of strains.

The second most affected site in Romania is Bucharest metropolitan area, a region with important worldwide connections. The first documented cases there were imported from different countries (Israel, Italy, Germany).

The phylogenetic analysis indicated that the formerly imported strains were spreading further at a local level through transmission networks observed as specific clusters in the tree. This is valid especially for Bucharest cluster 2, one that consists of sequences corresponding to patients who were epidemiologically linked. These sequences showcase one particular non-synonymous mutation, C16049T (T870I in Nsp12). In contrast, the sequences from Bucharest cluster 1 showed a higher degree of variability, indicating multiple introduction events. They were found to be closely related to the ones circulating in Switzerland, Italy, France and Turkey.

Our analysis showed that the SARS-CoV-2 strains circulating in Romania at the beginning of the epidemic have slightly higher non-synonymous than synonymous mutation frequencies suggesting viral host adaptation. It was reported that in the short time since the virus has adapted to use humans as hosts, different strains have emerged [25], characterized by specific mutational profile [26]. Most of the identified mutations were previously reported to occur in sequences circulating in Europe, while another (nt17746) was previously found to be exclusive in North America [26].

Some mutations have emerged as possible signatures for different outbreaks in Romania, thus A20268G is linked to north Romania (Suceava county), while sequences with G28881A, G28882A and G28883C are found in south-east Romania (BMA).

A mutation profile consisting in C3037T, C14408T, A23403G (mutation found in all the Romanian sequences), next to C241T was found to be specific to the Europe cluster and was initially associated with higher pathogenicity [27]. A subsequent study reported that A23403G (D614G in S) is becoming prevalent worldwide and associated with severe disease cases [28]. However, only one of the isolated strains (EPI_ISL_468152) was associated with severe disease; this strain had an additional mutation Y397C in Nsp4 protein. This mutation was found to be quite rare compared with other mutations: only 4 other cases of SARS-CoV-2 contain this mutation (1 sequence from Scotland and 3 sequences in England; the first mutation appeared in England with the following GISAID ID - EPI_ISL_423932), of which two are of B1 (EPI_ISL_458337, EPI_ISL_423932) and two of B2 (EPI_ISL_425922, EPI_ISL_421843) lineage, while ours is in the lineage B.1.1 according to the CoV-GLUE tool. At the time of the analysis and to our knowledge, this mutation is novel to scientific reports. The Nsp4 protein seems to play an important role in the replication process and mutations in the SARS CoV Nsp4 sites 120 and 121 were reported to affect virus replication [29]. A short overview on the structure of Nsp4 (Appendix A) shows that the Y397 residue is surrounded by three cysteines (C226, C256, C296) of which C226 and C296 are the closest to the tyrosine, but even if the other one is placed away, it resides on a long loop that acts like a hinge between two structural domains, therefore it is possible that during certain processes they loop to bring C256 in the proximity of Y397. It is plausible to presume that one of these cysteines would form a disulfide bridge if Y397C mutation occurs, therefore a fourth cysteine in that space could enhance the formation of the disulfide bridges in that region, which may affect the overall conformation of Nsp4. It is also known that Nsp4-Nsp3 interaction is critical for the viral replication in SARS-1 [29] but the interaction with Nsp6 is also important [30]. On the other hand, it is well known that tyrosines may be involved in various processes such as H-bond formation, phosphorylation, interface interaction, interactions with nucleotides, etc. However, additional data are needed in order to understand the impact of this change in Nsp4 on virus replication.

A new mutation, K489E in Nsp2 protein, was identified in one of the strains circulating in BMA in a patient with a mild disease form. The protein Nsp2 was found to be dispensable for SARS-CoV-1 replication [31], but it was demonstrated to interact with prohibitin 1 and 2 and perturb host cell environment [32]. Mutations in Nsp2 and Nsp3 proteins of SARS-CoV-2 were suggested to contribute to increased virulence as compared to SARS-CoV-1 [33]. A recent study reported a deletion in Nsp2 coding gene of SARS-CoV-2 strains circulating in France [34]. Although the unreported mutation (K489E) we have found in Nsp2 protein may not impact the virus replication, its interaction with the host mechanisms might influence the disease evolution.

There were differences in the mutation frequency when analyzing the sequences reported in this study in comparison with the other Romanian sequences available in GISAID, observed mainly in non-synonymous substitutions. This may be partly explained by different bioinformatic pipelines that were used to generate the viral WGS.

In conclusion, the phylogenetic analysis indicated that the SARS-CoV-2 Romanian epidemic was initiated by multiple introductions mainly from European countries followed by local transmissions. We have observed different subtype distribution between northern and southern Romania. This geographical subtype segregation was observed as a result of the travel restrictions instituted during the emergency state. The viral strains circulating in northern Romania were closely related with strains from Spain, Austria, Scotland and Russia, while the strains from southern Romania were highly similar to viruses from Switzerland, Italy, France and Turkey.

## Figures and Tables

**Figure 1 life-10-00152-f001:**
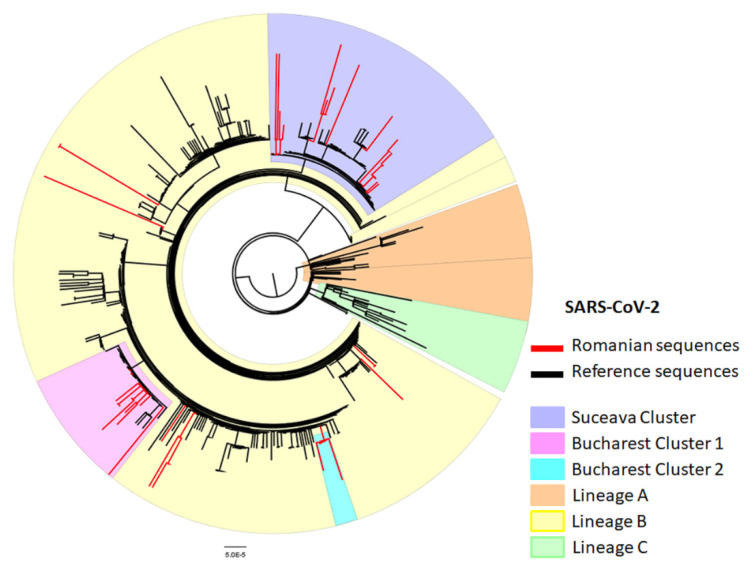
Phylogenetic analysis of Romanian SARS-CoV-2 sequences. Romanian sequences are represented as red branches, reference sequences are marked in black. SARS-CoV-2 type A cluster is highlighted in orange, type B in yellow and type C in green. Several Romanian clusters were distinctly marked in the tree: Suceava in purple, Bucharest cluster 1 in pink and Bucharest cluster 2 in cyan.

**Figure 2 life-10-00152-f002:**
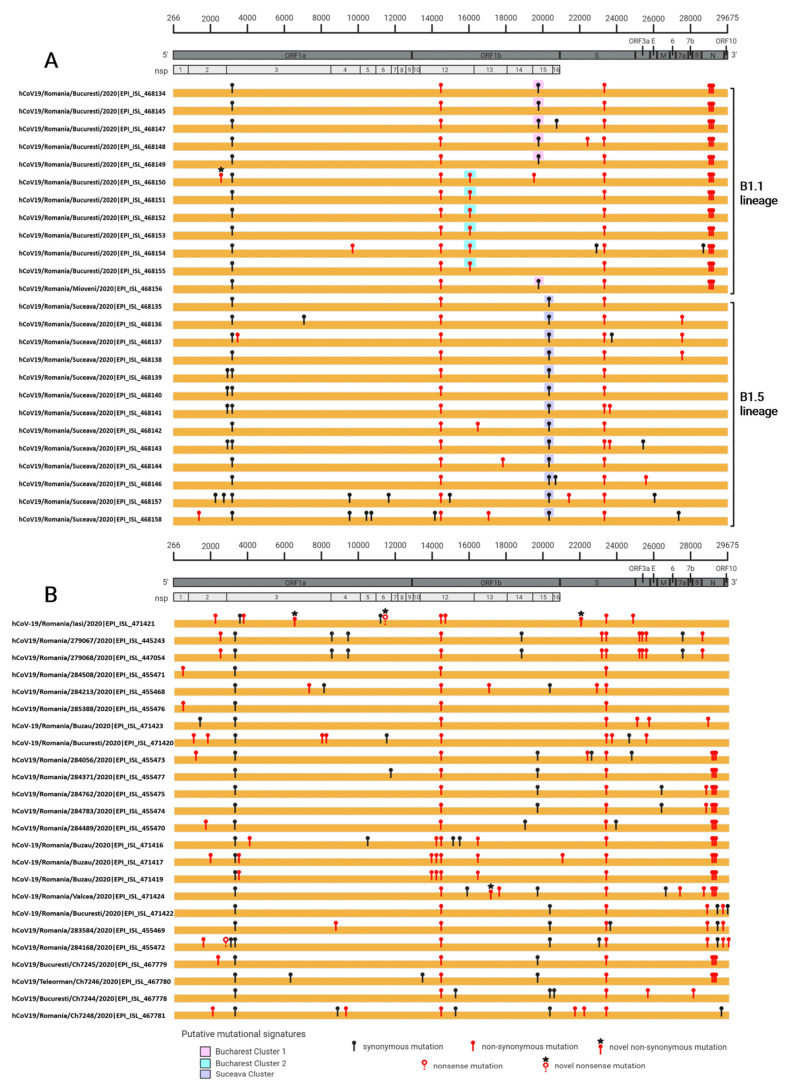
Mutational profile of the forty-nine Romanian SARS-CoV2 sequences. (**A**) Mutations present in the sequences generated in our study, depicted as either a black (synonymous mutation) or red pin (non-synonymous mutation), aligned using the 045512.2 NCBI reference sequence. Novel mutations were signposted with a black asterisk. Putative mutational signatures of observed clusters were highlighted as follows: pink for Bucharest Cluster 1, blue for Bucharest Cluster 2 and purple for Suceava Cluster. (**B**) The mutational profile of the other Romanian sequences retrieved from GISAID presented additional nonsense mutations (red pin with a hollowed center).

**Figure 3 life-10-00152-f003:**
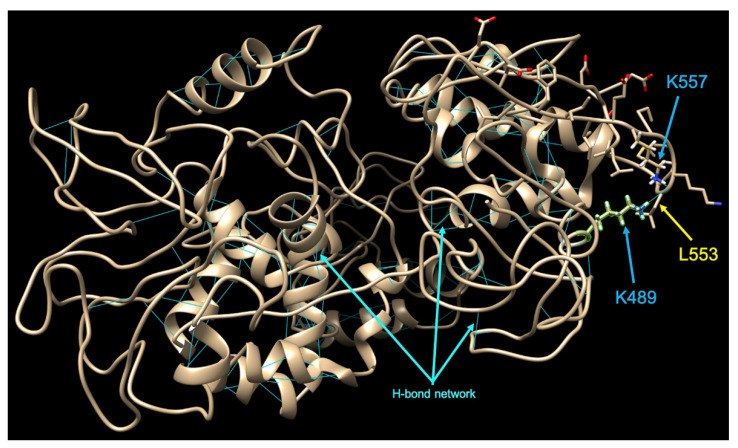
C-I-Tasser model of Nsp2—local representation of tertiary structure (using Chimera) around the novel K489E mutation. The possible effect of K489E mutation in Nsp2: with cyan lines are represented the H-bonds network in the Nsp2 model. The amino acids in the proximity of K489 are represented as sidechains. Hydrogens are represented on both K489 and K557 Lysins and the amino acids involved in H-bond interactions with K489 and L553 are shown.

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
