# Peer review of "Molecular Epidemiology Analysis of SARS-CoV-2 Strains Circulating in Romania during the First Months of the Pandemic"

_life, 2020, doi:10.3390/life10080152_

Round 1
Reviewer 1 Report
The authors present a comprehensive analysis of the main clusters of SARS-Cov2 in Romania
minor comments
The authors should state mean read depth and coverage after assembly
authors may emphasize also on the origins of the clusters from european or other isolates worldwide
Figure 2. needs improvement (black line)
Supplementary Figure 2 appears in negative colors
Author Response
The authors should state mean read depth and coverage after assembly
R: We thank the reviewer for this observation. We have modified it accordingly in the manuscript at line 113-114: “The de novo contigs representing the viral full genome had an average coverage of 182 reads with a minimum of 16 and the maximum of 543 reads.”
authors may emphasize also on the origins of the clusters from european or other isolates worldwide
R: Due to a number of limitations (low variability observed in a short time frame, no temporal information included), the phylogenetic analysis pinpointed rather the relatedness among sequences than the origin of the clusters.
Figure 2. needs improvement (black line)
R: Figure 2 was modified accordingly.
Supplementary Figure 2 appears in negative colors
R: We believe that Supplementary figure 1 is the one the reviewer was referring to and it was modified as suggested.
Reviewer 2 Report
The authors sequenced 25 respiratory samples from Romaina collected during the first two months after SARS-CoV-2 was identified in the country. They were interested in conducting a molecular epidemiological study of the origin and spread of the virus in Romania. All of the sequences were assigned to pangolin lineage B which led the authors to conclude that the origin of the virus in Romania was the result of multiple introduction events followed by community spread. They also observed different lineage B subtypes between northern and southern Romania. They also report a novel nonsynonymous mutation in one of the viral strains.
Overall the manuscript follows similar styles to other case studies that have assessed phylogenetic relatedness of SARS-CoV-2 in certain regions or countries. The authors focus on one particular novel nonsynonymous mutation for which they provide little functional evidence.
Major comments:
-The manuscript has very little to do with molecular evolution and is more of a molecular epidemiological report. The title does not fit the content of the manuscript.
-For the novel K489E mutation, there is little support that this mutation has functional importance. The authors acknowledge that there exists other mutations from EPI_ISL448303, EPI_ISL_448990, and EPI_ISL_463445 that also change the amino acid at position 489 but do not provide much support as to why these changes are unimportant.
-For the variability profile analysis, the Supplementary Figure 1 is missing axes and legends. It is difficult to tell what I am looking at. Since I am unable to interpret the figure, I cannot comment on the results from that analysis.
-The authors speculate about the functional importance about a mutation Y397C in the discussion without providing any experimental evidence.
Minor comments:
-The authors collected samples from different disease outcomes which they claim are "asymptomatic/mild, moderate, severe, critical". It is generally accepted that asymptomatics should form a separate group to mild cases and even pre-symptomatic cases. The authors don't state how these groups were defined nor do they use this information in any quantitative way.
-The authors should provide a table of the sample collection dates rather than stating the samples are from the first two months of the pandemic.
-In the introduction it is not widely accepted that pangolins were intermediate hosts for SARS-CoV-2
-Figures are blurry
-For Figure 1, the yellow color doesn't match the Lineage B legend. Also it is impossible to see where the 25 sequences that were part of this study are located in the tree. They are grouped with the other Romanian sequences already available in public databases.
- This paragraph should be omitted "This section may be divided by subheadings. It should provide a concise and precise description of the experimental results, their interpretation as well as the experimental conclusions that can be drawn."
-In the discussion, the authors mention that their sequences have higher nonsynonymous vs synonymous changes and suggest that this is evidence of "viral host adaptation" but the authors do not provide evidence showing that this is not a result of neutral evolution.
-In the discussion, the authors mention that the mutation frequency in their sequences is different from published Romanian sequences, which they suggest is a result of different bioinformatic pipelines. In the results, there seems to be no mention about this discrepancy.
-In the AIM, line 24 should be "performed a molecular epidemiology"
-In Conclusions line 38 should be "two months of the pandemic"
-line 46 should be "coronavirus infectious disease"
-line 129 Blast search should be all capitalized
-line 147 should be "All of the Romanian"
-line 178 should be "All of the analyzed"
-line 234 should be "All of the sequences"
-line 236 should be "all of the sequences"
Author Response
Major comments:
-The manuscript has very little to do with molecular evolution and is more of a molecular epidemiological report. The title does not fit the content of the manuscript.
R: We thank the reviewer for the suggestion; we have modified the title accordingly.
-For the novel K489E mutation, there is little support that this mutation has functional importance. The authors acknowledge that there exists other mutations from EPI_ISL448303, EPI_ISL_448990, and EPI_ISL_463445 that also change the amino acid at position 489 but do not provide much support as to why these changes are unimportant
R: Indeed, there is little support for the functional importance but we only speculated on a possible role when interactions essential to viral replication occur. By August the 10th 2020, we have noticed new modifications in the GISAID database, related to the mutations in this position. From the other 3 different mutations only the mutations to N and R are found in the database and the accession numbers are as follows: EPI_ISL_448990 for K489N mutation and EPI_ISL_478285 for K489R. From these mutations in the K489 position of NSP2, K489E has a higher probability to change the conformation in that place, favoring the salt-bridge formation.
We have therefore modified the text by replacing the sentence: “There are only three other mutations in the GISAID database at this particular position in Nsp2, but none to describe a K-to-E mutation. The other three mutations in this position are to D/N/R (EPI_ISL_448303, EPI_ISL_448990 and EPI_ISL_463445 respectively).” with: “There are only two other different mutations in the GISAID database at this particular position in Nsp2, but none to describe a K-to-E mutation. The other two mutations in this position are to N/R (EPI_ISL_448990 and EPI_ISL_478285 respectively).”
-For the variability profile analysis, the Supplementary Figure 1 is missing axes and legends. It is difficult to tell what I am looking at. Since I am unable to interpret the figure, I cannot comment on the results from that analysis.
R: We thank the reviewer for indicating this apparent omission. We have corrected the figure accordingly.
-The authors speculate about the functional importance about a mutation Y397C in the discussion without providing any experimental evidence.
R: There is, indeed, limited support for the functional importance of this mutation Y397C, but this is why we only invocated it as speculation. On the other hand we felt important to report this mutation which was identified in a viral strain from a patient with severe form of the disease.
Minor comments:
-The authors collected samples from different disease outcomes which they claim are "asymptomatic/mild, moderate, severe, critical". It is generally accepted that asymptomatics should form a separate group to mild cases and even pre-symptomatic cases. The authors don't state how these groups were defined nor do they use this information in any quantitative way.
R: Because of the relatively low sample size, we could not perform an in-depth analysis of sequence variation based on stage of disease. The phrase was modified according to the reviewer’s suggestion (line 88-89).
-The authors should provide a table of the sample collection dates rather than stating the samples are from the first two months of the pandemic.
R: the collection date information will be available when accessing the GISAID accession numbers.
-In the introduction it is not widely accepted that pangolins were intermediate hosts for SARS-CoV-2
R: the following sentence has been modified accordingly: “Early molecular epidemiology studies on SARS-CoV-2 revealed the initial zoonotic transmission of this virus from bats (the main reservoir of coronaviruses) and possibly other intermediate hosts (e.g. Malaya pangolins) [10,11].”
-Figures are blurry
R: the figures (2, supplementary 1) were modified to increase visibility, hopefully all the figures can be seen in the right format.
-For Figure 1, the yellow color doesn't match the Lineage B legend. Also it is impossible to see where the 25 sequences that were part of this study are located in the tree. They are grouped with the other Romanian sequences already available in public databases.
R: the lineage B is highlighted in light yellow (border box in the legend is indeed different from the interior, just to be more visible). To simplify the visualization of sequences in the tree, we have used red branches for all the Romanian sequences.
- This paragraph should be omitted "This section may be divided by subheadings. It should provide a concise and precise description of the experimental results, their interpretation as well as the experimental conclusions that can be drawn."
R: Thank you for the observation; we have removed this from the revised text.
-In the discussion, the authors mention that their sequences have higher nonsynonymous vs synonymous changes and suggest that this is evidence of "viral host adaptation" but the authors do not provide evidence showing that this is not a result of neutral evolution.
R: A higher non-synonymous mutation rate suggests that they were actively selected as opposed to the synonymous mutations which are passively selected (e.g. through bottle-neck events).
-In the discussion, the authors mention that the mutation frequency in their sequences is different from published Romanian sequences, which they suggest is a result of different bioinformatic pipelines. In the results, there seems to be no mention about this discrepancy.
R: We thank the reviewer for this observation. We have added the following phrase in the Results section at lines 212-215: “We noticed a difference in the non-synonymous mutational frequencies when analyzing the viral genomes sequenced in this study and other Romanian sequences available in GISAID platform: ~25% versus ~75% of the total non-synonymous mutations accounted.”
-In the AIM, line 24 should be "performed a molecular epidemiology"
R: addressed
-In Conclusions line 38 should be "two months of the pandemic"
R: modified accordingly
-line 46 should be "coronavirus infectious disease"
R: modified accordingly
-line 129 Blast search should be all capitalized
R: modified accordingly
-line 147 should be "All of the Romanian"
R: revised as suggested
-line 178 should be "All of the analyzed"
R: revised as suggested
-line 234 should be "All of the sequences"
R: revised as suggested
-line 236 should be "all of the sequences"
R: revised as suggested